# Important Lessons on Long-Term Stability of Amino Acids in Stored Dried Blood Spots

**DOI:** 10.3390/ijns9030034

**Published:** 2023-06-21

**Authors:** Allysa M. Dijkstra, Pim de Blaauw, Willemijn J. van Rijt, Hanneke Renting, Ronald G. H. J. Maatman, Francjan J. van Spronsen, Rose E. Maase, Peter C. J. I. Schielen, Terry G. J. Derks, M. Rebecca Heiner-Fokkema

**Affiliations:** 1Section of Metabolic Diseases, Beatrix Children’s Hospital, University Medical Center Groningen, University of Groningen, 9700 RB Groningen, The Netherlands; a.m.dijkstra@umcg.nl (A.M.D.); w.j.van.rijt@umcg.nl (W.J.v.R.); f.j.van.spronsen@umcg.nl (F.J.v.S.); t.g.j.derks@umcg.nl (T.G.J.D.); 2Laboratory of Metabolic Diseases, Department of Laboratory Medicine, University Medical Center Groningen, University of Groningen, 9700 RB Groningen, The Netherlands; p.de.blaauw@umcg.nl (P.d.B.); hanneke-r@hotmail.com (H.R.); r.g.h.j.maatman@umcg.nl (R.G.H.J.M.); 3Centre for Health Protection, Dutch National Institute for Public Health and the Environment, 3721 MA Bilthoven, The Netherlands; rose.maase@rivm.nl; 4Centre for Population Screening, Dutch National Institute for Public Health and the Environment, 3721 MA Bilthoven, The Netherlands; peter.schielen@rivm.nl

**Keywords:** amino acids, dried blood spots, inborn metabolic diseases, metabolite stability, neonatal blood spot screening, tandem mass spectrometry

## Abstract

Residual heel prick Dried Blood Spots (DBS) are valuable samples for retrospective investigation of inborn metabolic diseases (IMD) and biomarker analyses. Because many metabolites suffer time-dependent decay, we investigated the five-year stability of amino acids (AA) in residual heel prick DBS. In 2019/2020, we analyzed 23 AAs in 2170 residual heel prick DBS from the Dutch neonatal screening program, stored from 2013–2017 (one year at +4 °C and four years at room temperature), using liquid chromatography mass-spectrometry. Stability was assessed by AA changes over the five years. Hydroxyproline could not be measured accurately and was not further assessed. Concentrations of 19 out of the remaining 22 AAs degraded significantly, ranked from most to least stable: aspartate, isoleucine, proline, valine, leucine, tyrosine, alanine, phenylalanine, threonine, citrulline, glutamate, serine, ornithine, glycine, asparagine, lysine, taurine, tryptophan and glutamine. Arginine, histidine and methionine concentrations were below the limit of detection and were likely to have been degraded within the first year of storage. AAs in residual heel prick DBS stored at room temperature are subject to substantial degradation, which may cause incorrect interpretation of test results for retrospective biomarker studies and IMD diagnostics. Therefore, retrospective analysis of heel prick blood should be done in comparison to similarly stored heel prick blood from controls.

## 1. Introduction

Blood spots dried on filter paper cards, i.e., dried blood spots (DBS), are valuable samples commonly used in the field of inborn metabolic diseases (IMD) for screening purposes, such as the population newborn screening (NBS), and monitoring purposes, e.g., in regular patient care [1,2,3]. In The Netherlands, 18 IMDs are currently included in the NBS program, among which are disorders of mitochondrial fatty acid oxidation, disorders of organic acid metabolism and disorders of amino acid (AA) metabolism [4]. These disorders can be detected rapidly by analyses of targeted AA and acyl-carnitines (AC) using tandem mass spectrometry (MS/MS) [5].

After analysis, residual blood from neonatal screening or regular patient care can be stored, in this way providing samples for retrospective evaluation of biomarkers, particularly for rare diseases, [6] or retrospective diagnostic measurements, for example, in case of post-mortem investigations [7].

In the Netherlands, DBS cards from the NBS program are stored centrally by the Reference Laboratory for NBS at the Dutch National Institute of Public Health and the Environment (RIVM) for a maximum period of five years after laboratory analyses, one year for quality assurance purposes and four years for anonymized scientific research, if parents do not object [4]. Internationally, there is no consensus between screening facilities on storage times and conditions for samples from the NBS programs. Therefore, storage times and conditions of collected DBS vary not only worldwide, but even within countries [8]. To be able to correctly interpret results from retrospective laboratory analyses, information on stability of biomarkers in stored DBS is essential.

In a recent study, we performed AC and AA analyses in residual heel prick blood to study the prevalence of IMD in children who died in early childhood (i.e., <5 years of age, as DBS are only stored for this period of time). The storage conditions (one year at +4 °C followed by four years of storage at room temperature (RT)), strongly affected the five-year stability of carnitine and most AC, complicating interpretation of results [9].

Short-term stability of AA, most often in plasma, has been described earlier [10,11,12,13,14]. However, only a few articles describe long-term stability of AA in DBS [15,16,17,18]. While most previous studies highlight the conditions that affect long-term metabolite stability (especially high temperatures, high humidity and long storage times), they often lack a description of clinical implications resulting from these findings. Moreover, these articles only describe stability of a selection of AA (+/−10). To provide information on long-term stability of AA, we report the investigation of 23 AAs in a cohort of 2170 residual heel prick DBS from the NBS program, stored from 2013 to 2017. Similar to our previous AC stability study, we describe how metabolite instability can complicate the interpretation of AA biomarkers in retrospectively analyzed residual heel prick blood for IMD diagnosis.

## 2. Materials and Methods

### 2.1. Study Design

This study is connected to a larger research project that assesses the prevalence of IMDs, by measurements of AC and AA in residual heel prick blood from the Dutch NBS program, in children who died in early childhood (to be published). Instability of carnitine and AC in this residual heel prick blood complicated interpretation of the analytical data [9]. Because the same is to be expected for AA concentrations, here we investigate AA stability in residual heel prick blood.

In the Dutch NBS program, heel prick DBSs are obtained between 72 and 168 h after birth. These blood spots are dried at RT, and subsequently transported and analyzed (at RT) in one of five regional screening laboratories. In our study, residual heel prick blood obtained in the period 2013–2017 was used. Until 2018, residual heel prick blood was stored at the regional screening laboratory for quality assurance purposes at +4 °C for one year following the NBS analyses, and, if there was no parental objection, for another four years at RT at RIVM for anonymized scientific research. Residual heel prick blood was stored in bundles of 200–300 cards in sealed plastic bags, in cardboard boxes (20–30 bags per box), without conditioning of temperature and humidity.

At the start of the study (October 2018), we retrieved 1 DBS of 1570 residual heel prick DBS cards from children who died in early childhood and of 600 anonymized residual heel prick DBS cards with normal screening results (*n* = 120 DBS-cards per storage year cohort), yielding a total of 2170 samples. This resulted in five storage year cohorts (2013, 2014, 2015, 2016 and 2017). The large number of samples per storage year cohort permits the nonparametric determination of reference intervals [19,20]. Upon retrieval from RIVM, we first performed a visual inspection and smell test on the DBS to rule out significant effects of mold or bacterial contamination that could have induced amino acid decay independent of time (and temperature). No mold was inspected in the samples. Until analyses, DBS cards were kept in sealed bags with freshly added silica sachets, stored in a freezer at −20 °C in the laboratory of Metabolic Diseases, University Medical Center Groningen (UMCG) in Groningen, the Netherlands. A timeline of the storage conditions of the residual heel prick blood used in this study is shown in Table 1.

### 2.2. Ethical Statement

The Medical Ethical Committee of the UMCG confirmed that the Medical Research Involving Human Subjects Act did not apply for this study (METc code 2016/694). The Research Committee on Neonatal Screening (WONHS in Dutch) of the RIVM and the UMCG approved, and both granted a waiver of consent, since this study concerned the use of anonymized samples. Residual heel prick blood from neonates with parents who objected to use of the residual heel prick blood for anonymous scientific research were excluded.

### 2.3. DBS Amino Acid Analysis

Ammonium formate, methanol and acetonitrile were purchased from Biosolve (Valkenswaard, the Netherlands). Physiological AA standards (acidics, neutrals, basics), formic acid and L-citrulline-4,4,5,5-D4 were purchased from Merck KGaA (Darmstadt, Germany). The internal standard (I.S.) mix (metabolomics amino acids mix standard) was purchased from Cambridge Isotope Laboratories (Cambridge, MA, USA). Milli-Q^®^ Ultrapure water quality was used (Merck KgaA, Darmstadt, Germany). Quality Control (QC) DBS samples (ClinChek^®^, Whole blood control for Acylcarnitine and Amino Acids, QC low and QC high) were obtained from Recipe Chemicals (München, Germany). A representative QC sample control was produced in the laboratory by spotting heparin-anticoagulated blood from a healthy adult volunteer onto Whatman™ 903 filter paper (Eastern Business Forms, Mauldin, SC, USA). After drying at RT, DBSs were stored in sealed plastic bags with silica sachets at −20 °C at the UMCG.

Prior to the analysis, samples were randomized and divided into 27 equal batches to prevent bias from possible analytical inter-batch variations. Each batch consisted of a calibration curve, approximately 90 samples and 4 internal quality control (QC) samples. Batches were analyzed between September 2019 and June 2020. Of each DBS sample, a 3.2 mm blood spot disc was punched in a deep 96-well plate and 150 µL of methanol containing the internal standard (I.S.) mix (50 µM for each I.S.) was added. The plate was sealed with a pierceable cap mat, and AAs were extracted by vortexing for 30 minutes at 600 RPM. The extracted samples were transferred to a clean 96-well plate and placed in the autosampler for analysis. For the calibration curves, 2.4 µL of each calibration standard (7 levels 10–1000 µM) was transferred in an Eppendorf cup, and 150 µL of methanol containing the I.S. mix was added. Samples were subsequently vortexed. The solutions were transferred to a 1.5 mL glass vial with insert and placed in the cooled autosampler for analysis

A Hydrophilic Interaction Liquid Chromatography (HILIC) method combined with tandem mass spectrometry was developed to detect and quantify AA. The method was based on previously published methods with minor modifications [21]. In alphabetical order, the following 23 AA were analyzed: alanine (Ala); arginine (Arg); asparagine (Asn); aspartate (Asp); citrulline (Cit); glutamine (Gln); glutamate (Glu); glycine (Gly); histidine (His) hydroxyproline (Hyp); isoleucine (Ile); leucine (Leu); lysine (Lys); methionine (Met); ornithine (Orn); phenylalanine (Phe); proline (Pro); serine (Ser); taurine (Tau); threonine (Thr); tryptophan (Trp); tyrosine (Tyr) and valine (Val).

A 3 µL extracted sample was injected in the UHPLC-MS/MS system. Chromatographic separation was achieved using an ACQUITY UPLC BEH Amide Column (130 Å, 1.7 µm, 2.1 mm × 100 mm; Waters, Milford, MA, USA) and a Shimadzu Nexera UHPLC system (Kyoto, Japan). The temperature of the analytical column was maintained at 40 °C. Mobile phase A consisted of 0.1 v/v% formic acid in 10 mM ammonium-formate dissolved in 100% MilliQ-water. Mobile phase B consisted of 0.1 v/v% formic acid in 10 mM ammonium-formate dissolved in 95 v/v% acetonitrile/MilliQ-water. The applied gradient was (%mobile phase B): 0–1.5 min. 90%; 1.5–6.0 min. 75%; 6.0–8.0 min. 72%; 8.0–8.1 min. 50%; 8.1–10 min. 50%; 10.0–10.1 min. 90%; 10.0–14.0 min. 90%. The flow rate was 0.4 mL/min. A Sciex (Framingham, MA, USA) 4500 Qtrap mass spectrometer was used to monitor the transitions. Electrospray ionization (ESI) in positive mode was used. The electrospray voltage was set to 5500 V. Nitrogen was used as collision, carrier and curtain gas. The capillary temperature was 750 °C. Sciex Analyst^®^MD 1.6.2 software was used to acquire the data and for processing data we used Sciex Multiquant^®^MD 3.0.3.

A correction factor for the measured AA was used, which was based on the correction factor for Phe and Tyr that was previously established by van Vliet et al [22]. This correction factor corrects for matrix effects and DBS punch volume, enabling use of liquid calibrators instead of DBS calibrators. This approach increases the analytical precision of the calibration curves, as DBS measurements have inherently high analytical imprecision, thereby improving the between-run analytical precision. The correction factor was applied in the volume of calibration samples used for sample preparation, i.e., 2.4 μL, and resulted in good recoveries for DBS Phe (97–107%) and DBS Tyr (93–99%) in the 2019–2022 ERNDIM quantitative external quality control schemes in DBS. The assayed commercial DBS QC samples also showed no bias compared to the established ranges for most AA levels, with a few exceptions showing only minor biases (Appendix A). No such comparison could be made for the other AA, but we assumed that this factor is also representative for other DBS AA concentrations.

Information about the retention times, applied I.S. for each AA, *m*/*z* transitions, MS settings of the compounds, inter- and intra-assay precisions, linearity, Coefficients of variation (CV) and LOD/LOQ can be found in Appendix A.

### 2.4. Statistical Analysis

Batch results were merged, and a dataset was constructed in Microsoft Excel. CVs were calculated to determine the precision of the method per AA from the data of the QC samples, by dividing the standard deviation by the means of the analyzed samples. Any measures outside the 1.5 × IQR (interquartile range) criterion were considered outliers and were excluded from further analyses [23]. After outlier removal, samples of controls and deceased children were analyzed together.

First, a multivariate overview principal component analysis (PCA) of all the data was performed to visualize multidimensional data and explore the complete dataset. The Hotelling’s T2 statistic at *p* = 0.95 is shown to visually present the (multivariate) outliers in the PCA. The distribution of AA concentrations and their molar ratios per storage year cohort were assessed. The Jonckheere’s trend test was applied to study significant trends upon storage duration, using a *p*-value of <0.05 as statistically significant. When significant, mean changes and their 95% confidence intervals were calculated.

For AA showing significant changes upon long-term storage, regression analyses were used to define trends. Annual decay rates from linear trends were estimated from the slope of the trend line equations. The five-year percent decays were calculated from the estimated decay rates and the median AA concentrations in 2017. Five-year percent decays for other trend types were calculated from the difference in the median concentrations in 2017 and 2013 and analyzed for significant differences using a student’s *t*-test. 

To indicate until what timepoint, under the given storage conditions AA can be interpreted, we calculated the time (in years) after which the reference change value (RCVa) was reached. The RCVa is the maximum acceptable percentile deviation based on the analytical variance (CVa) and a factor for 95% uncertainty. The RCVa was calculated as follows: RCVa = 1.96 × √2 × (mean CVa). Mean CVa values were used that were calculated from the DBS, low QC and high QC CVs (Appendix A).

Data analysis was performed using SIMCA Software, version 15.0.2 (Umetrics, Umea, Sweden), Microsoft Excel with the Analyse-it add-in, version 4.81.6, and IBM SPSS Statistics for Windows, version 28 (IBM Corporation, Armonk, NY, USA). To create graphs, GraphPad Prism, version 9.0 (GraphPad Software, La Jolla, CA, USA) was used.

### 2.5. CLIR Search Strategy

To investigate the effect of storage on the measured AA concentrations in residual heel prick blood and the consequence for retrospective IMD diagnosis based on analyses from these stored samples, we compared our data to data from the ‘Collaborative Laboratory Integrated Reports’ (CLIR—https://clir.mayo.edu, accessed on 20 February 2023) Productivity Tools.

CLIR is a Web-based application consisting of an interactive database of NBS results from multiple sites. It provides high throughput-post-analytical interpretive tools to improve NBS performance. The data in CLIR include cumulative reference intervals for biomarkers, including AA concentrations and ratios, and the replacement of analyte cut-off values with an integrated scoring based on the degree of overlap between reference ranges and condition-specific disease ranges.

To identify disease informative AA, the CLIR Productivity Tools were systematically searched: ‘plot by marker’, ‘AA’, ‘only AA disorders’. All AA were independently searched for disease informative markers. To identify informative disease related ratios, the CLIR Productivity Tools were searched for ‘Productivity tools’, ‘plot by condition’, ‘AA disorders’, ‘select markers—ratios to AA’. All AA disorders were independently searched for informative ratios to AA with a maximum of 10% overlap with the reference range.

## 3. Results

### 3.1. Validation

A total of 2170 residual heel prick DBSs were analyzed: 600 from healthy controls and 1570 from deceased children. Appendix A show the limits of detection and quantification (LOD & LOQ), and intra- and inter-assay precision of 23 AAs in a freshly prepared DBS QC and in two commercial DBS QC samples (low and high). Acceptance criteria for these parameters were not specified prior to the validation because it was uncertain which variation would be expected. Appendix A shows that all 23 AAs can be measured in DBS, as their mean concentrations were above the LOD and LOQ. Arg, Asp, Gln, Lys and Orn had high inter-assay CVs (defined as >25%). Intra-assay CVs were also high for Asp and Hyp. Hyp had high CV in all QC samples, whereas this was less consistent for these other AAs. Results for Hyp were therefore omitted for further investigations. High CV in an AA other than Hyp may be partly explained by the different matrices of the pooled and the commercial QC samples, as CV were lower for the pooled QC sample for Arg, Asp, Gln and Lys. It may also be the result of the instability of these AA in QC samples, which is further strengthened by the changes seen in the differences between the intra-assay and inter-assay means of Arg, Asp, Gln, Lys and Orn (Appendix A) Because of the high CVs, assessment of these AAs in DBS may be less reliable, yet clear decay trends could be observed; we therefore chose to present these AAs as well as the other AAs, but marked them grey in all figures and tables.

### 3.2. AA Stability

Figure 1 shows the PCA scatter plot, including all AAs. Each sample is depicted as a colored dot, the colors representing the year of collection. The first principal component of the score plot suggests an annual shift in the underlying data, explaining 36.9% of variation.

Individual outlier AA measurements were removed. Remaining AA measurements from the same residual heel prick DBS, if normal, were kept for further analyses. In total, 4652 (9.3%) measurements of the 49,910 individual AA measurements (2170 samples × 23 AAs) were removed based on the 1.5 × IQR criterion. Changes in remaining AA concentrations, depicted as annual mean concentrations (annual decay), are shown as a series of boxplots in Figure 2. A summary of the findings on the annual decays is shown in Table 2. Jonckheere’s trend test showed significant negative trends upon long-term storage for 19 out of 22 AAs; ranked from most to least stable over the total storage period: Asp, Ile, Pro, Val, Leu, Tyr, Ala, Phe, Thr, Cit, Glu, Ser, Orn, Gly, Asn, Lys, Tau, Trp and Gln. Asp remained stable for at least four years (between 2017–2014 + additional storage in our center before analyses) and decreased significantly thereafter. Ala, Ile, Pro and Tyr remained stable for at least 1 year (between 2017–2016 + additional storage in our center) and significantly decreased thereafter. All other AAs had significantly decreased already within one year of storage. Trend analyses of Arg, His and Met could not be interpreted, as a significant number of the measurements for these AAs were below the laboratory detection limit. Using regression analysis, trends for all AAs were defined as linear, except for Gln (exponential decay). The largest absolute and percentile-wise decrease in AA concentrations between 2013 and 2017, apart from Arg, His and Met, was seen in Gln, with an estimated five-year decay of 92%. For the remaining AAs, the average annual percentile decay in the concentrations of AA ranged between 4 and 25%.

### 3.3. AA Ratio Stability

Jonckheere’s trend tests were also performed for the AA ratios regarded as informative (CLIR) for the diagnosis and monitoring of various IMDs [3]. Results are shown as a series of box plots in Figure 3. Thirty AA ratios are informative for various IMDs (Table 3). Thirteen ratios included the AA Arg or Met and could not be interpreted because of instability of these AAs. Of the remaining 17 informative ratios, seven (Cit/Phe, Glu/Cit, Ala/Cit, Orn/Cit, Xle/Tyr, Phe/Tyr and Tyr/Pro) remained stable during the five years of storage. For these ratios, individual percentual decay rates of the AA were comparable (e.g., 6% and 7% for Phe/Tyr). The remaining ten ratios changed significantly over five years of storage.

### 3.4. The Impact of AA Instability on Retrospective Investigations

The impact of long-term storage of residual heel prick blood at RT on the interpretation of AA and AA ratios for validation studies or retrospective identification of IMDs is shown in Table 3. The CLIR Productivity Tools have been used as the source to determine which AA and AA ratios can be regarded as informative. Metabolite instability increases the likelihood of incorrect interpretation of AA biomarkers and AA ratios for IMDs in residual heel prick blood stored at RT. Twenty out of twenty-three (20/23) AA concentrations significantly decreased upon long term storage. For most of the associated disorders, this increases the risk of false-negative test conclusions for the identification of IMDs when applying cut-off values employed in the routine NBS. Ten out of seventeen (10/17) AA ratios were unstable at prolonged storage, which increases the incidence of false-negative or false-positive test results.

## 4. Discussion

Use of stored residual heel prick blood is important for retrospective diagnosis of IMDs and biomarker analyses. However, some metabolites in these residual heel prick DBS suffer time-dependent decay, which impacts their measured concentration, and can drastically affect interpretation of results. Several studies previously reported the effects of short-term storage in plasma and DBS [10,11,12,13,14]. Less has been written about instability of AAs in long-term stored DBS, nor its effect on retrospective identification of IMDs and biomarker studies. In this study, we investigated AA stability in DBS following storage of up to approximately five years and elaborated on the effects of retrospective use of these samples.

Alterations in metabolite concentrations, including AAs, are accelerated by suboptimal pre-analysis and storage conditions, such as increased storage time, temperature, humidity, and sunlight exposure, which can account for a large number of diagnostic misinterpretations in clinical chemistry laboratories [24]. In our study, residual heel prick DBS cards were stored at 4 °C for one year, at RT for four years, and for some months at −20 °C before analysis. Our results show that, under these combined conditions, measured concentrations of 19 out of 22 investigated AA significantly decreased during five years of storage; ranking from most to least stable: Asp, Ile, Pro, Val, Leu, Tyr, Ala, Phe, Thr, Cit, Glu, Ser, Orn, Gly, Asn, Lys, Tau, Trp and Gln. In addition, a considerable impact was seen for most molar ratios, while only seven molar ratios remained stable.

The AAs Ala, Ile, Pro and Tyr were stable for at least one year (between 2017–2016 + additional storage at the UMCG before analyses) and significantly decreased thereafter. It is uncertain whether this decrease in AA concentrations after one year was attributed to the shift from storage at 4 °C to RT, yet this is difficult to assess without comparison with samples stored at RT from the start. Except for Asp (stable for four years), all other AA concentrations significantly decreased within one year of storage. The course of Arg, His and Met concentrations in our study could not be interpreted, as nearly all concentrations were below the laboratory detection limit within one year of storage. Arg was likely to have been converted to Orn by arginase. Hannemann et al. showed that Arg concentrations drastically decreased in high humidity conditions at RT (−96.5% within one month) [25]. Significant increases in Orn were not seen in this study. However, since samples were already stored for some period before analysis in our center, it could be that initial increases in Orn were missed. Degradation of Met concentrations was probably due to oxidation of an S-methyl thioether group that is susceptible to oxidation, especially under high temperature conditions [14]. This was likely to have been degraded because of the chemical instability of the heterocyclic indole and imidazole sidechain in its biochemical structure [14]. The storage conditions in our study (RT, uncontrolled humidity) are likely explain the complete degradation of Arg, His and Met. Earlier studies also showed that Met is one of the least stable AAs in DBS [10,17]. Meanwhile, His concentrations in DBS have rarely been investigated.

Discrepancies between previous publications illustrate the complexity in characterizing and defining the stability of metabolites in residual heel prick blood. Our results agree with a number of smaller scale studies on long-term stability of AAs in DBS [15,17], but differ from the results of some others: For example, Strnadova et al. found no significant changes in Val concentrations, while we observed a significant decrease with an annual decay of 5% [17]. This may be explained by Val being especially sensitive to storage temperature and humidity. In line with this hypothesis, Adam et al. found 6% and 9% loss of Val concentrations at low and high humidity during 30 days of storage at 37 °C, respectively [11]. Gln was one of the least stable AAs in our study, alongside Arg, His and Met. The substantial instability of Gln in our samples is in line with results from Prentice et al. and is most likely caused by glutaminase activity [15]. In plasma, instability of Gln, which is easily converted to Glu by glutaminase, is well known [26]. Moreover, at higher temperatures, Gln is also known to reduce to pyroglutamate by cyclization, in which case Glu does not increase in relation to the decrease in Gln [27]. Our results showed a non-significant increase in Glu during the first year of storage, but a significant decrease during the remaining years. This may suggest that Gln was converted to pyroglutamate instead of Glu, and that Glu is itself unstable.

Metabolite instability as illustrated by our study may result in incorrect interpretation of the results of retrospective analysis of stored residual heel prick blood. This in turn may lead to false-positive or false-negative assignment of IMDs. Therefore, knowledge of AA stability is essential. Correct retrospective identification of IMDs is dependent on understanding of metabolite stability, initial metabolite concentrations and the significance of the metabolite decay assessed against its disease specific target cut-off values. For many of the DBS AAs, the disease target range (specifically between the 1st–10th percentile and the 90th–100th percentile of the disorder range) overlaps with the healthy newborn reference intervals [3]. This is observed for Tyr concentrations evaluated in the diagnosis of Tyrosinemia Type I and for Val concentrations for Maple Sirup Urine Disease, for example. Depending on where the cut-off value is set, this overlap may result in a false assignment of the result. When employing stored residual blood in the retrospective diagnosis of IMDs, this effect is further aggravated by metabolite instability.

Although retrospective diagnosis for many IMDs may be challenging using stored residual heel prick blood, some diseases may still be diagnosed effectively even after years of DBS storage at non-optimal storage conditions, particularly when AA ratios can be applied. Phe, for example, showed annual decay rates of 6% (30% at five years). With this rate of decay, after five years, initial concentrations of Phe should have been >172 or >267 µmol/L to exceed the 120–200 µmol/L cut-off values that are adhered to in many countries [3,28]. For most cases, diagnosis may therefore still be feasible, as classic phenylketonuria patients often have Phe concentrations much higher at the NBS. Because the Phe/Tyr ratio remained stable, Phe/Tyr can be assessed in addition to minimize the possibility of false-negative results when using stored DBS for diagnosis of PKU. Caution is warranted, however, since Golbahar et al. showed that, under extreme temperatures (45 °C), Phe and Tyr displayed different decline rates (20% vs. 50% respectively in 1 day), thereby also affecting the Phe/Tyr ratio [10].

In practice, retrospective analysis of residual heel prick blood spot for the purpose of diagnosis or scientific research is mostly qualitative and should therefore always be done in comparison to residual heel prick blood from controls that are stored for a similar period of time under similar storage conditions. Significant differences between the AA concentrations in these controls and investigated ‘suspected’ samples should then be used to retrospectively confirm diagnosis of IMDs.

The results of our study emphasize the importance of obtaining background information on sample storage conditions. While it is evident that decay in AA concentrations cannot be generalized and should be evaluated individually for the analytes that are investigated, Prentice et al. and Michopoulos et al. showed that drastic changes in AA and AC concentrations in stored DBS can be substantially reduced or prevented by DBS storage at −20 °C or −80 °C [15,29]. Therefore, storage at −20 °C or −80 °C seems like a good way to improve AA stability. Before recommendations for storage at −20 °C (or even −80 °C) can be made, further research should investigate AA stability in residual heel prick blood upon long-term storage at −20 °C or −80 °C and the results compared to those of this study. New storage conditions at the RIVM allow for such a comparison. Since 2018, the storage procedure for NBS samples in the Netherlands at the RIVM has been optimized: residual heel prick blood is now stored in custom made racks, at 4 °C by the regional laboratories for a period of 3 months. Thereafter, the residual heel prick blood is transported to RIVM and stored at −20 °C (uncontrolled humidity) for up to 1 or 5 years. All residual heel prick blood is stored for one year and from 2023 parents of neonates must give their permission for storage of the residual heel prick blood of their child for up to five years.

As humidity is also a strong determinant of instability of metabolites in DBS, maintaining a humidity <30% seems preferable, in addition to storage at −20 °C or −80 °C [10,24]. This is also important during transport of the DBS, where temperature and humidity should be kept low and stable as much as possible [9,11,30].

Alongside the findings, there are a number of limitations, Samples were anonymized and, therefore, initial AA concentrations are unavailable, restricting calculation of true within spot decays. Secondly, the initial NBS analysis method (flow-injection MS/MS) measures only a limited number of AA, does not separate isomers, and methods have changed in 2018. Moreover, a correction factor, which was established previously for Phe and Tyr [22], was used to calculate DBS AA concentrations based on liquid calibrators in our LC-MS/MS method, assuming similar recoveries and matrix effects for all other AAs. This assumption seemed correct for most AAs, as mean concentrations of the commercial QC samples fell within the range of those AAs supplied by the vendor (Appendix A). All these factors complicate the comparison between our and the NBS screening methods. For assessment of stability, the relative change in concentrations is, however, most relevant, not the bias to the actual DBS concentration. AA changes were therefore compared to median AA concentrations in 2017 instead of initial concentrations, which was possible given the large sample size. However, because of this, variations of the estimated AA instabilities were relatively large.

Next, earlier studies suggest a more rapid degradation during the first months of storage at RT that eventually stabilizes [31,32]. This short-term instability could not be investigated in our study, but we estimate the decay of the AA during longer-term storage.

In addition, samples of controls and deceased children were analyzed together, because: (1) diagnosis is not important for this study, as the focus was to understand the effect of storage on AA concentrations in residual heel prick blood, (2) deceased children, with expected IMDs, can still serve as control for any AA that is not affected by this IMD, (3) outliers, consistent with possible IMDs, were removed, essentially making all DBS samples as suitable ‘control’ samples and (4) analyzing the controls alone drastically lowered the sample size and power, which, upon comparison, negatively affected the linearity of the results (Appendix A).

Lastly, because all samples employed in this study were anonymized, we had no access to similarly stored NBS samples of patients diagnosed with various IMDs to validate the claimed impact on detection of IMDs, and instead used information from the CLIR productivity tools to draw our conclusions.

## 5. Conclusions

AA profiles in DBS stored at RT are subject to substantial metabolite degradation. For this reason, retrospective analysis of residual heel prick blood spot for the purpose of diagnosis or scientific research should be carried out with caution, and always in comparison to residual heel prick blood from controls.

## Figures and Tables

**Figure 1 IJNS-09-00034-f001:**
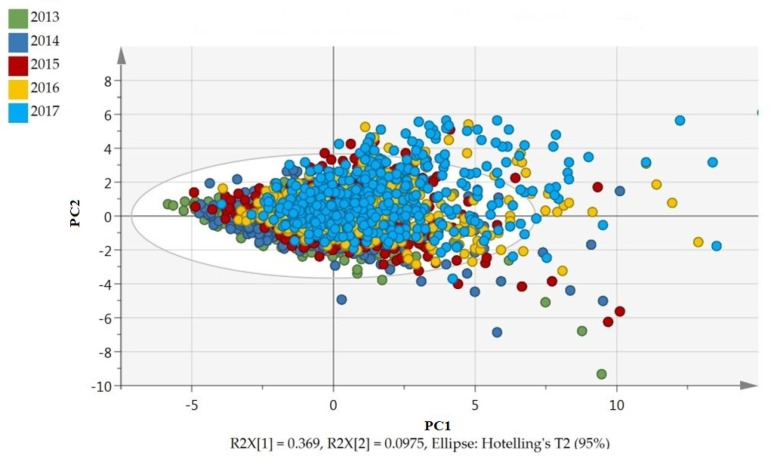
Principle Component Analysis (PCA) of all samples, showing an annual shift in DBS AA concentrations. Data were colored according to cohort year: each variable has equal variance. (Figure in color.). PC1 or PC2 = principle component 1 or 2, R2X[1] or R2X[2] = percentage variance explained by PC1 or PC2.

**Figure 2 IJNS-09-00034-f002:**
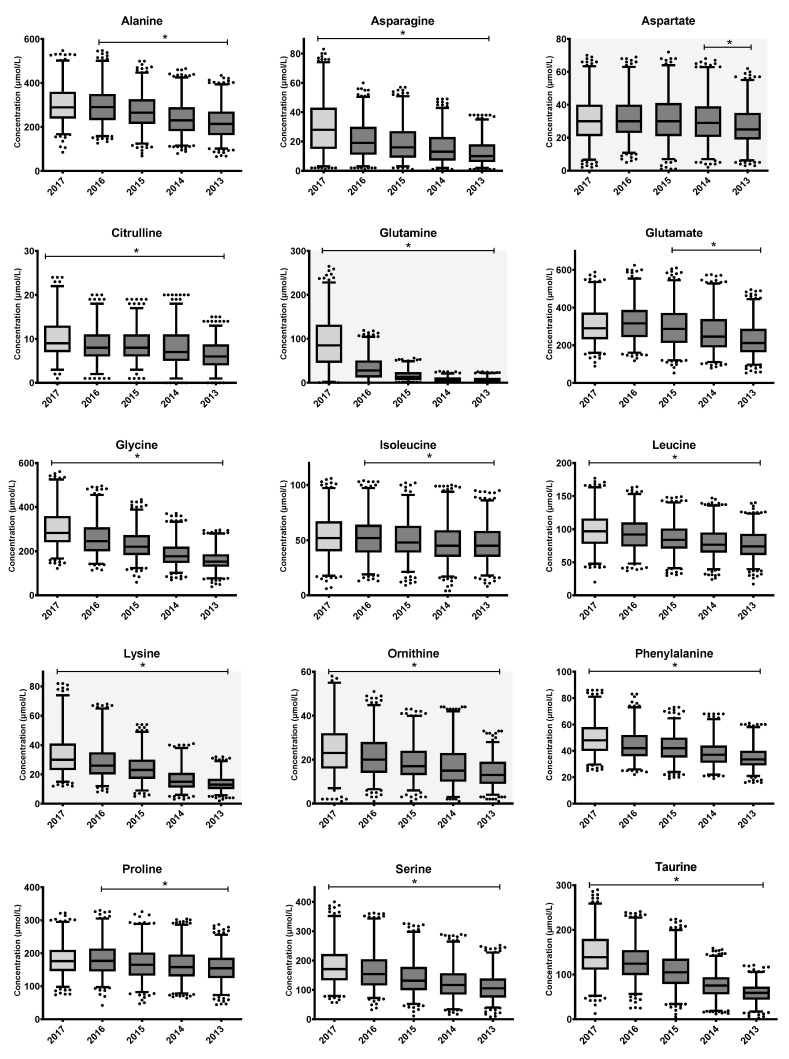
Changes in AA concentrations upon long-term storage at room temperature. The boxplots represent the first quartile, median and third quartile. The whiskers extend to the 2.5th and 97.5th percentiles. Individual dots represent non-extreme outliers. Asterisks ‘*’ above boxplots represent statistically significant trends in the concentration upon storage duration, as determined by Jonckheere’s trend test. Samples from 2017 were stored at +4 °C (light grey box), while other samples transferred to storage at RT after one year (dark grey boxes). Trend analyses of Arg, His and Met could not be interpreted, because many measurements were below the detection limit. Assessment of Asp, Arg, Orn, Gln and Lys (indicated with grey background) was possibly less reliable due to high CVs.

**Figure 3 IJNS-09-00034-f003:**
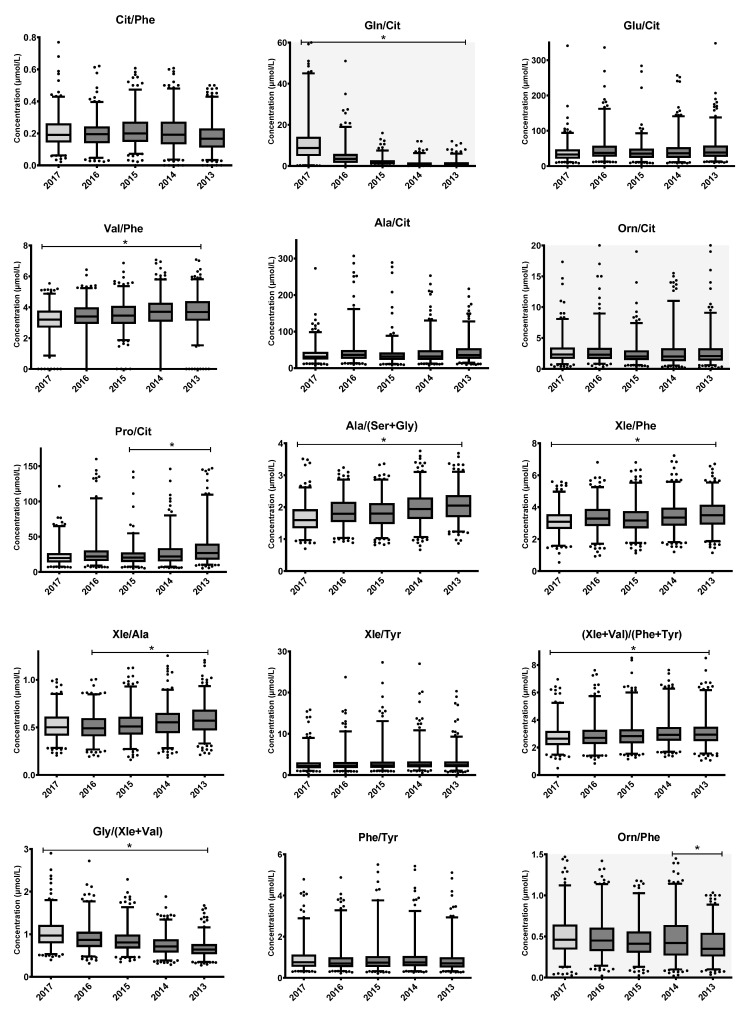
Changes in AA ratios used in IMD diagnoses and monitoring upon long-term storage of DBS at RT. Boxplots represent the first quartile, median and third quartile. The whiskers extend to the 2.5th and 97.5th percentiles. Samples from 2017 were stored at +4 °C (light grey box), while other samples were transferred to storage at RT after one year (dark grey boxes). Individual dots represent non-extreme outliers. Asterisks ‘*’ above boxplots represent statistically significant trends in the concentration upon storage duration, as determined by Jonckheere’s trend test. Trend analyses of the ratios consisting of Arg, His or Met could not be interpreted because of (short-term) instability of these AAs. Assessment of Orn and Gln was possibly less reliable due to high CVs.

**Table 1 IJNS-09-00034-t001:** Total storage time and storage conditions of DBS employed in this study.

Group	2013	2014	2015	2016	2017	2018		2019		2020		Storage Times (yrs)
2013	Collection and storage at +4 °C						Transport to UMCG + storage at −20 °C		AA measurements			4.75–5.75
2014		Collection and storage at +4 °C		Storage at room temperature								3.75–4.75
2015			Collection and storage at +4 °C									2.75–3.75
2016				Collection and storage at +4 °C								1.75–2.75
2017					Collection and storage at +4 °C							0.75–1.75

Residual heel prick blood was collected from the cohort years 2013–2017. All DBS cards were stored for one year at +4 °C followed by storage at RT for another four years. This is displayed in the table in relation to the year of heel prick blood collection. For example, in the cohort of 2013, some samples are taken in January 2013 and stored at +4 °C until January 2014, while others might be collected in December 2013 and stored until December of the next year. Total storage time is always one year at +4 °C and 4 years at RT. Upon arrival at the UMCG, DBS cards were stored at −20 °C until analysis.

**Table 2 IJNS-09-00034-t002:** Average annual decay (+/−95%CI level) 2013–2017.

Amino Acid	Model of Decay	Mean Annual Decay	
		Absolute decay (µM)	Percentile decay (%)
Alanine	Stable between 2017–2016. Linear decay from 2016–2013	22 (14–30)	7 (5–10)
Arginine	Cannot be interpreted	-	-
Asparagine	Linear decay	4 (2–6)	13 (7–20)
Aspartate	Stable between 2017–2014. Linear decay from 2014–2013	1 (0–2)	3 (0–6)
Citrulline	Linear decay	1 (0.3–1.4)	8 (3–14)
Glutamine	Exponential decay		2017–2016: 63% 2017–2015: 82% 2017–2014: 91% 2017–2013 92%
Glutamate	Non-significant increase 2017–2016 (5%). Linear decay from 2016–2013	30 (20–40)	9 (6–12)
Glycine	Linear decay	36 (31–41)	12 (10–13)
Histidine	Cannot be interpreted	-	-
Isoleucine	Stable between 2017–2016. Linear decay from 2016–2013	2 (1.2–2.5)	4 (2–5)
Leucine	Linear decay	6 (5–7)	6 (5–7)
Lysine	Linear decay	5 (4–6)	15 (12–19)
Methionine	Cannot be interpreted	-	-
Ornithine	Linear decay	3 (2–3)	10 (8–13)
Phenylalanine	Linear decay	4 (3–4)	7 (6–8)
Proline	Stable between 2017–2016. Linear decay from 2016–2013	7 (3–10)	4 (2–6)
Serine	Linear decay	18 (15–21)	10 (8–12)
Taurine	Linear decay	23 (18–27)	15 (12–19)
Threonine	Linear decay	9 (8–10)	8 (7–9)
Tryptophan	Linear decay	4 (3–5)	25 (18–32)
Tyrosine	Stable between 2017–2016. Linear decay from 2016–2013	4 (2–6)	6 (3–9)
Valine	Linear decay	7 (5–10)	5 (3–6)

The model of decay (linear or other) for each AA and both the mean absolute (µM) and percentual annual decay (%), including their 95% confidence intervals are presented. Trends for Arg, His and Met could not be interpreted because many measurements were below the detection limit. Assessment of Asp, Arg, Orn, Gln and Lys was possibly less reliable due to high CVs (indicated in light grey).

**Table 3 IJNS-09-00034-t003:** Impact of metabolite instability on interpretation of amino acid concentrations.

		Effect of Individual AA on Retrospective Assessment	Assessment of Parameters/Disorders			Effect of AA Disease Ratios on Retrospective Assessment	
Parameter *	Disorder	Retrospective Analysis of IMDs	RCVa	Annual Percentile Decay	RCVa Reached	Metabolite Ratio	Retrospective Analysis of IMDs
		Risk Category	(%)	(%)	(years)		Risk Category
**Arg**	**ARG**	False-negative	56.5	- ***	- ***	Arg/Orn, Arg/Phe, Arg/Ala	Cannot be interpreted
						Cit/Arg **(low)**	Cannot be interpreted
**Cit (low)**	**OTC/CPS**	False-positive	31.9	8	4.0	No informative AA disease ratios	-
	**NAGS, OAT**	False-positive	31.9	8	4.0	No informative AA disease ratios	-
**Cit**	**CIT-I**	False-negative	31.9	8	4.0	Cit/Phe	none
						Cit/Arg, Met/Cit **(low)**	-
						**(Low)**: Ala/Cit, Glu/Cit, Orn/Cit	none
						Pro/Cit **(low)**	False-negative
						Gln/Cit **(low)**	False-positive
	**CIT-II**	False-negative	31.9	8	4.0	Cit/Phe	none
						Pro/Cit **(low)**	False-negative
						Gln/Cit **(low)**	False-positive
	**PC**	False-negative	31.9	8	4.0	Cit/Phe	none
						Met/Cit **(low)**	-
						**(low)**: Ala/Cit, Glu/Cit, Orn/Cit	none
						Pro/Cit **(low)**	False-negative
	**ASA**	False-negative	31.9	8	4.0	**(low)**: Cit/Phe, Ala/Cit	none
						Pro/Cit **(low)**	False-negative
						Orn/Cit **(low)**	False-positive
						Met/Cit **(low)**	-
**Gln (low)**	**CIT-II**	False-positive	53.7	Exponential decay (Table 2)	<1	*See CIT – CIT-II*	
**Glu (low)**	**PC**	False-positive	21.6	*9*	2.4	*See CIT-PC*	
**Gly (low)**	**3PGDH**	False-positive	23.0	12	2.2	Ala/(Ser + Gly)	False-positive
**Gly**	**NKHG**	False-negative	23.0	12	2.2	No informative AA disease ratios	
**Xle ****	**MSUD**	False negative	19.11	6	3.2	Val/Phe, Xle/Phe, Xle/Ala, Xle/Tyr	False-positive
						Met/Xle **(low)**	-
						(Ile + Leu + Val)/(Phe + Tyr)	False-positive
						Gly/(Ile + Leu + Val)	False-negative
**Xle (low)**	**BCKDK**	False-positive	19.11	6	3.2	**(low)**: Val/Phe, Xle/Phe, Xle/Ala, (Xle + Val)/(Phe + Tyr)	False-negative
**Met (low)**	**RMD**	False-positive	18.6	**- *****	**- *****	Met/Phe **(low)**	Cannot be interpreted
**Met**	**HCY**	False negative	18.6	**- *****	**- *****	Met/Ala, Met/Xle, Met/Phe, Met/Cit, Met/Val, Met/Gly, Met/Pro	Cannot be interpreted
	**H-MET**		18.6	**- *****	**- *****	Met/Ala, Met/Phe, Met/Cit, Met/Val, Met/Gly, Met/Tyr, Met/Pro, Met/Xle	Cannot be interpreted
**Orn (low)**	**SSADH**	False-positive	55.1	10	>5	No informative AA disease ratios	
**Phe**	**PKU, H-PHE, BIOPT (BS/Reg)**	False-negative	20.8	7	3.0	Phe/Tyr, Cit/Phe **(low)**	none
						**(low)**: Val/Phe, Xle/Phe, (Ile + Leu + Val)/(Phe + Tyr) (PKU)	False-negative
						**(low)**: Met/Phe, Arg/Phe	-
						Orn/Phe **(low)**	False-positive
**Pro**	**H-PRO**	False-negative	15.8	4	4.0	Pro/Cit	False-positive
						Orn/Pro **(low)**	False-positive
**Tyr**	**TTI, TTII, TTIII**	False-negative	15.8	6	2.6	Tyr/Pro (TT2)	False-negative
						Phe/Tyr, Xle/Tyr **(low)**	none
						Met/Tyr **(low)**	-
						(Ile + Leu + Val)/(Phe + Tyr) **(low)**	False-negative
**Val**	**MSUD**	False-negative	19.11	*5*	*3.8*	See Xle	
**Val (low)**	**BCKDK**	False-positive	19.11	*5*	*3.8*	See Xle (low)	

* ‘Low’ is added if low values of the AA are indicative of the disease. If no indication is added, this indicates elevated values. ** Xle represents the combination of the isomers Leu, Ile and allo-Ile that are not separated using flow-injection MS/MS methods. *** Arg, His and Met concentrations were below LOQ in all samples in our study, rendering evaluation of ratios including these AA not feasible. Table 3 shows the consequences associated with the interpretation of retrospectively analyzed AA and AA ratios in DBS upon long-term storage at RT for the detection of IMDs, based on the CLIR-database. Abbreviations (in alphabetical order): ASA, arginino-succinic acidemia (#207900); ARG, argininemia (OMIM, #207800); BCKDK, Brached-Chain Keto Acid Dehydrogenase Kinas Deficiency (#614923); BIOPT (BS), disorders of biopterin biosynthesis (#261640); CB1, cobalamin (homocystinuria-megaloblastic anemia, complementation type (#236270) CIT-I, citrullinemia type I (#215700); CIT-II, citrullinemia type II (#605814, #603471); CPS, carbamyl-phosphate synthase deficiency (#237300); HCY, hyper-homo-cysteinemia (#603174); H-PHE, hyperphenylalaninemia (#261600); H-PRO, hyper-prolinemia (#239500); MSUD, maple syrup urine disease (#248600); H-MET, hyper-methioninemias (#250850); NAGS, N-acetyl-glutamate Synthase Deficiency (#237310); NKHG, nonketotic hyperglycinemia (#605899); OTC, ornithine trans-carbamylase deficiency (#300461); PKU, phenylketonuria (#261600); PC, pyruvate carboxylase deficiency (#266150); RMD, Remethylation defects (homocystinuria-megaloblastic anemia, CblG #250940 and CblE #236270 complementation types and (N)5,10-methylenetetrahydrofolate reductase (MTHFR) deficiency #236250); SSADH, Succinic Semialdehyde Dehydrogenase Deficiency (#271980); TTI, tyrosinemia type I (#276700); TTIII, tyrosinemia type III (#276710); TTII, tyrosinemia type II (#276600); 3PGDH deficiency, Phosphoglycerate dehydrogenase deficiency (#601815). The AAs: Ala, Asn, Asp, His, Lys, Ser, Tau, Thr and Trp were not indicative disease markers and were therefore excluded from the table. The disorders: NKHG, TT1, NH, OAT, OTC/CPS and NAGS had no informative AA disease ratios. Assessment of Arg, Orn, and Gln was possibly less reliable due to high CVs (indicated in light grey).

## Data Availability

The data presented in this study are available on request from the corresponding author. The data are not publicly available due to privacy restrictions.

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
