# Peer review of "Important Lessons on Long-Term Stability of Amino Acids in Stored Dried Blood Spots"

_2409-515X, 2023, doi:10.3390/ijns9030034_

Round 1

Author Response

Response to reviewer 1:

Dear reviewer,

First of all we would like to thank you for your continued interest, your additional comments and your questions concerning our manuscript entitled ‘Important lessons on long-term stability of amino acids in stored dried blood spots’.  We have addressed your comments and questions, which we further elaborate on below, and enclosed our revised manuscript. We think the comments and questions contributed significantly to our manuscript.

Yours sincerely,

Also on behalf of the other authors,

Rebecca Heiner-Fokkema

Comments:

Comment/question 1 Addressed the fact that we did not specify if we considered the possibility of mold or bacterial contamination on the investigated DBS during the storage period. The reviewer asked whether we did any tests to rule out mold (such as PCR) and commented that we should otherwise state in our manuscript that a visual inspection/smell test was done to rule out mold.

We did not perform a PCR to investigate mold or bacterial contamination on our specimen. We visually inspected our samples for the presence of mold, which was not identified. Therefore, we added a sentence to the manuscript that we did a visual inspection and smell test to rule out significant effects of mold or bacterial contamination as alternative cause of altered amino acid concentrations, unrelated to time or temperature.

Comment/question 2 commented on the high commercial QC levels of five amino acids that were found in our results, and asked to both elaborate on how these values compare to the CoA from the vender and what the reason for such large variation in amino acid measurements can be.

Thank you for this comment. The high CV are indeed remarkable, and also high compared to the pool DBS sample. We think that it may be explained by the different matrices of the pooled and the commercial QC samples. It may also be the result of the instability of these AA in QC samples, which is further strengthened by the changes seen in the differences between the intra-assay and inter-assay means of these affected AA, namely Arg, Asp, Gln, Lys and Orn (Supplemental table 2.A). We also compared these CV’s to the concentrations of the commercial QC samples supplied by the vendor. Only Hyp had high CV in all QC samples, whereas this was less consistent for these other AA. Results for Hyp were therefore omitted for further investigations. As trends in the instability of the other amino acids were very clear, and our goals of this manuscript is to assess relative change instead of the bias to the actual DBS concentrations, we decided to present these amino acids. However, to clarify more to the reader which amino acids had poor performance, we have changed the legends of the figures and tables to address this point.

Reviewer 2 Report

This study investigated the decay of each amino acid in DBS over time and how this decay affects the detention of IMD, and is expected to have a significant academic and social impact.

I believe that this manuscript qualifies for publication in this journal, but some revisions are necessary before publication.

major comments

  • In this study, the authors applied the previously reported plasma-DBS correction factors for Phe and Tyr to all amino acid measurements. I believe that the scientific rationale for this should be provided. It is known that amino acid values from filter paper are different from plasma amino acid values. I cannot see the point of applying this kind of processing. Considering the theme of this article, I believe that it does not make sense to go to the trouble of converting the measured values by applying coefficients for which the scientific basis is not clear. 

  • It would be valuable information for readers to have a compilation of estimated years after sample collection under the current storage conditions (1-year storage at 4°C followed by subsequent room temperature storage), which is believed to allow for accurate assessment for each disease or parameter.

minor comments

  • Amino acid analysis is a vital component of this study; however, there is a lack of detailed description regarding the analysis method. While descriptions related to LC-MS/MS analysis using HILIC columns can be found, there is insufficient information regarding sample pre-treatment (extraction time, extraction method, etc.). A concise description of these methods is necessary.
  • Concerning Figure 2, as the storage conditions vary considerably between the samples from 2017 and subsequent ones, it is recommended to make visual enhancements for improved understanding.
  • Typo:  

    3.2 AA Stability

    Third line from the top of the second paragraph: 49.910 49,910

  •  

Author Response

Response to reviewer 2:

Dear reviewer,

First of all we would like to thank you for your continued interest, your additional comments and your questions concerning our manuscript entitled ‘Important lessons on long-term stability of amino acids in stored dried blood spots’.  We have addressed your comments and questions, which we further elaborate on below, and enclosed our revised manuscript. We think the comments and questions contributed significantly to our manuscript.

Yours sincerely,

Also on behalf of the other authors,

Rebecca Heiner-Fokkema

Comment/question 1 Stated that we should provide scientific rationale for our choice to use a plasma-DBS correction factor, based on Phe and Tyr concentrations, for alle investigated amino acids.

The reviewer addresses a very important point. Calibration of methods in DBS is difficult, as good calibration samples, preferably in the same matrix (DBS), are often lacking and difficult to make yourselves, especially when many components with different chemical compositions are studied at the same time. We chose a pragmatic approach, using a correction factor that corrects for matrix effects and DBS punch volume, allowing us to use liquid calibrators. The applied correction factor showed to result in good recoveries for DBS Phe (97-107%) and DBS Tyr (93-99%) in the 2019-2022 ERNDIM quantitative external quality control schemes in DBS. The assayed commercial DBS QC samples from this study also showed no bias compared to the established ranges for most AA levels, with a few exceptions showing only minor biases (Supplemental materials table 2.A). No such comparison could be made for the other AA, but we assumed that this factor is also representative for other DBS AA concentrations. This approach also increases the analytical precision of the calibration curves, as DBS measurements have inherently high analytical imprecision, thereby improving the between-run analytical precision (1). We added additional information in the text to further elaborate on this point (see pg 4). The accuracy of the concentrations nevertheless remains an interesting point of discussion. In this manuscript, we investigated changes upon storage. A slight bias in the DBS concentration, though constant(!), is not problematic when investigating instability of metabolites, as only relative concentration differences are studied. We therefore feel confident that the instability results presented in our study are accurate.

  1. VAN VLIET, Kimber, et al. Dried blood spot versus venous blood sampling for phenylalanine and tyrosine. Orphanet journal of rare diseases, 2020, 15.1: 1-8.

Comment/question 2 Stated it would be a valuable addition to the readers to add a compilation up to what time point in years, certain diseases can be assessed accurately by measurement of informative amino acids based on the investigated storage conditions.

We agree with this point and added three columns in table 3 (impact of metabolite instability on interpretation of amino acid concentrations) with 1) the RCVa (reference change value; the maximum acceptable percentile deviation based on the analytical variance (CVa) and a factor of uncertainty), 2) the annual percentile decay and 3) after what time in years, the RCVa is reached (thus till what year accurate assessment of amino acids, based on the reference change value, can be made). Adding these 3 columns helps indicate how long accurate assessment of amino acids and associated disorders can be made based on the given storage conditions.

Comment/question 3 stated that we should describe our used analytical method for quantification of amino acids with more detail, especially with regards to sample pre-treatment (extraction time and method etc.)

We thank the reviewer for this comment. The detailed information was previously submitted as supplemental materials to save length from the article. We now moved some of this information to the main text to give the methods some more detail on our analytical process (see page 4/5).

Comment/question 4 Stated that we should provide some visual enhancements for figure 2 to improve understanding of the storage times.

We now changed the color of the first box plot (2017) to indicate that samples from 2017 were stored at 4 degrees (indicated as light grey box), while samples from other years transferred to storage at room temperature (indicated in dark grey).

Comment/question 5 Highlighted a typo in our manuscript.

We are grateful for the reviewer’s vigilant assessment of our work and we corrected this typo in the manuscript.